# Evaluation of 8% Capsaicin Patches in Chemotherapy-Induced Peripheral Neuropathy: A Retrospective Study in a Comprehensive Cancer Center

**DOI:** 10.3390/cancers15020349

**Published:** 2023-01-05

**Authors:** Florent Bienfait, Arthur Julienne, Sabrina Jubier-Hamon, Valerie Seegers, Thierry Delorme, Virginie Jaoul, Yves-Marie Pluchon, Nathalie Lebrec, Denis Dupoiron

**Affiliations:** 1Anaesthesiology and Pain Department, Institut de Cancérologie de l’Ouest, 49100 Angers, France; arthur.julienne@ico.unicancer.fr (A.J.); sabrina.jubier-hamon@ico.unicancer.fr (S.J.-H.); thierry.delorme@ico.unicancer.fr (T.D.); virginie.jaoul@ico.unicancer.fr (V.J.); nathalie.lebrec@ico.unicancer.fr (N.L.); denis.dupoiron@ico.unicancer.fr (D.D.); 2Biometrics Department, Institut de Cancérologie de l’Ouest, 49100 Angers, France; valerie.seegers@ico.unicancer.fr; 3Pain Management Consultation Center, Centre Hospitalier Départemental Vendée, 85925 La Roche-sur-Yon, France; yves-marie.pluchon@ght85.fr

**Keywords:** capsaicin, painful chemotherapy-induced peripheral neuropathy, CIPN, cancer painful sequelae

## Abstract

**Simple Summary:**

Chemotherapy-induced peripheral neuropathy (CIPN) is often painful, arising during or after the end of oncological treatments. The high-concentration capsaicin patch (HCCP) is recommended in second line for its treatment but based on low-powered studies. The objective of this retrospective real-world-data study was to evaluate efficacy and tolerability of HCCP applications in CIPN. Our study demonstrated an important or complete pain relief for 33.2% of the applications, corresponding to 43.9% patients. We found a significative difference in efficacy depending on the responsible chemotherapy. The efficacy was significatively different depending on the analgesic treatment line for HCCP. The efficacy of HCCP was significatively higher starting the third application. HCCPs were mainly responsible for local adverse events. HCCP applications in painful CIPN induce important pain relief with a global satisfying tolerability.

**Abstract:**

*Introduction:* Chemotherapy-induced peripheral neuropathy (CIPN) is often painful and can arise during or after the end of oncological treatments. They are mostly induced by platinum salts, taxanes, and immunotherapies. Their incidence is estimated between 19 and 85%. They can require a chemotherapy dose reduction or early termination. The European Society for Medical Oncology (ESMO) recommends high-concentration capsaicin patch (HCCP) in second line for the treatment of painful CIPN. This treatment induces a significative pain relief but only shown by low-powered studies. The objective of this study was to evaluate efficacy and tolerability of HCCP applications in CIPN. *Methods:* This monocentric observational retrospective real-world-data study of the CERCAN cohort took place in the Western Cancer Institute’s Anaesthesiology and Pain Department at Angers, France. Independent pain physicians completed the CGIC (Clinician Global Impression of Change) for each patient who benefited from HCCP applications for painful CIPN starting from 1 January 2014 to 22 December 2021, based on the collected data after every patch application. *Results:* A total of 57 patients (80.7% women) was treated with HCCP for painful CIPN, and 184 applications were realized, consisting of 296 sessions. CGIC found an important or complete pain relief for 61 applications (33.2%, corresponding to 43.9% patients). We found less efficacy for platinum-salts-induced CIPN compared to others (*p* = 0.0238). The efficacy was significatively higher for repeated applications when HCCP was used in second line compared to third line (*p* = 0.018). The efficacy of HCCP was significatively higher starting the third application (*p* = 0.0334). HCCPs were mainly responsible for local adverse events found in 66.6% patients (65.1% burning or painful sensation, 21.1% erythema). *Conclusion:* HCCP applications in painful CIPN induce an important pain relief with a global satisfying tolerability.

## 1. Introduction

The considerable progress made in oncologic treatments in recent decades combined with earlier diagnoses have made it possible to lessen mortality for most cancer types [1]. In these situations of prolonged survival, the long-term side effects of oncological treatments can be observed more often. This includes adverse events occurring during the treatment as well as post-chemotherapy, post-surgery, or post-radiotherapy sequelae. Painful sequelae are the most frequently reported side effects [2]. This pain often alters the patients’ quality of life and has a significant emotional impact.

Neuropathic pain is defined as “pain caused by a lesion or disease of the somatosensory nervous system” [3]. Some chemotherapies can induce painful neuropathies by altering the sensitive nerves. Platinum salts (cisplatin, carboplatin, and oxaliplatin) and taxanes (paclitaxel and docetaxel) are the most commonly involved in CIPN [4]. Some immunotherapies such as nivolumab or pembrolizumab also seem to induce painful neuropathies [5].

The prevalence of CIPN is 60% at one month after the end of the treatment and 30% at 3 months, including all types and total doses combined of chemotherapy [6]. The incidence varies from 19% to 85% depending on the studies [6]. Sometimes, neuropathic pain aggravation is observed weeks after the end of chemotherapy in platinum-salts-induced painful neuropathies due to coasting phenomenon [7]. Specific predictive factors have been identified, such as the patient’s age, the total dose, the co-administration of other neurotoxic chemotherapies, and some pre-existing conditions such as vitamin B deficiency, diabetes, or alcoholism [8]. CIPN are length-dependent neuropathies, mostly affecting the feet and hands and spreading upwards depending on their severity. The pathophysiological mechanisms of CIPN are still unclear, but chemotherapy-induced painful neuropathies seem multifactorial, resulting, for example, in taxanes from axonopathies, neurotoxic effects on myelin sheets, and soma alterations in the dorsal root ganglia, which are not protected by the blood–brain barrier [9,10]. Regarding platinum salts, which have poor cancer cells selectivity, when entering the sensitive neurons soma in the dorsal root ganglia, they seem to bind nuclear and mitochondrial DNA, provoking a decrease in transcription and mitochondrial replication and massive oxidation of the cellular component, resulting in neuronal death by apoptosis [11]. The neuronal soma alterations can lead to a lack of sensitivity or even dysautonomic disorders [12]. They can induce proprioceptive disorders, which result in postural deficits [13]. The occurrence of such adverse events can lead to a reduction in chemotherapy doses or even protocol discontinuation [14,15]. This neuropathic condition can result in chronicization, with persistence of the sensory peripheral neuropathy, while the motor peripheral neuropathy usually improves over time [16]. This risk of chronicity has been evaluated as being as high as 30% to 40% [17,18,19]. Moreover, CIPN have been shown to result in increased healthcare costs and sometimes impact work loss [20]. It thus seems crucial to detect and manage such adverse events as early as possible.

In Europe, guidelines have recently been updated by the ESMO regarding the treatment of CIPN [21]. Among the usual pharmacological treatments used in neuropathic pain, the only one that showed significant efficacy in a large randomized clinical trial is duloxetine, an SSRI antidepressant, mostly effective on CIPN mediated by platinum salts [22]. Since 2009, use of a topical treatment based on high-concentration capsaicin patches (HCCP) has been approved by the European Medicines Agency (EMEA) in peripheral neuropathy in non-diabetic patients and post-herpetic neuralgia by the U.S. Food and Drug administration (FDA) [23]. Its approval for use was then extended in the following years to use in Europe in diabetic patients in 2015 [24]. Its efficacy in localized peripheral neuropathic pain is observed regardless of the etiology [25]. Thus, this treatment is recommended as a second line in peripheral neuropathic pain [26]. An interesting point with this treatment is that it fills a gap in medication prescriptions for neuropathic pain by offering an alternative to the use of lidocaine patches for neuropathic pain, which do not have a marketing authorization (MA). Derived from red chili pepper, capsaicin is a TRPV1 receptor agonist. These receptors are mostly situated on the free nerve endings of unmyelinated thermoalgic C-fibers. The C-fibers transmit pain messages to the superficial Rexed’s laminae of the medulla’s dorsal horn and then to the spinothalamic tract. Prolonged activation of these cutaneous TRPV1 receptors triggers a calcium-mediated desensitization mechanism in the TRPV1 receptors and local C-fibers, leading to local elevation of the pain threshold for an extended time [27]. Significant pain relief has been demonstrated on localized neuropathic pain, such as postherpetic neuralgia, post-HIV neuralgia, and diabetic neuropathy [28,29]. The galenic formulation of a patch makes possible very low and brief systemic exposure, thus greatly limiting the occurrence of systemic adverse events [30]. However, its most reported adverse event, the burning sensation, which occurs during and/or after application, may have provoked some reluctance in its use. Nevertheless, in oncological practice, a local treatment with limited systemic exposure has numerous benefits thanks to the lessened risk of the interaction with oncological treatments such as hormonotherapy used in breast cancer [31]. Studies evaluating the efficacy of capsaicin patches in CIPN are scarce and of limited statistical power but are in favor of significant pain relief, as shown in Table 1 [32,33,34,35].

In this context, a retrospective monocentric cohort named CERCAN was established at the Institut de Cancérologie de l’Ouest (ICO) in Angers, France, a comprehensive cancer center, following approval by the local ethics committee. This cohort included all the patients who benefited from HCCP applications between 2014 and 2018 for peripheral neuropathic pain. Of the 996 patients included, a sub-cohort analysis recently showed significant relief from neuropathic pain resulting from cancer treatments in breast cancer [36]. The main objective of this study was to assess the efficacy of HCCP in the CERCAN sub cohort consisting of patients with CIPN. The secondary objectives were to evaluate the tolerability of HCCP applications, the time lapse before efficacy, and the duration of their efficacy in the same sub cohort.

## 2. Materials and Methods

### 2.1. Study Design and Patient Selection

This was a monocentric observational retrospective real-world-data study of the CERCAN cohort. It took place in the ICO Anaesthesiology and Pain Department in Angers, France, a comprehensive cancer center. The CERCAN cohort was constituted by including all the patients who had benefited from HCCP applications at the ICO between 1 January 2014 and 31 December 2018 for neuropathic pain whether or not it was related to their cancer. In this study, the population studied corresponded to the CERCAN sub cohort consisting of patients treated for CIPN. Concerning the patients that had already been included in the previous CERCAN publication concerning neuropathic pain in breast cancer, 33 patients suffered from CIPN and met the inclusion criteria and were thus included [36]. Exclusion criteria included an HCCP application starting before 31 December 2013 and HCCP applications for neuropathic pain not related to CIPN.

### 2.2. Study Treatment

HCCP was administrated at the ICO outpatient unit by nurses following a decision by the patient’s usual pain physician. The usual physician identified the area to be treated as well as the number of applications and the time interval between applications when applicable. The patch application duration usually ranged from 30 min for palms and soles to one hour for the back of hands and feet, as prescribed by the pain physician in charge. We do not use local anesthetics patches to prevent induced pain, but cooling packs were used when necessary. An application was defined as a HCCP treatment on the entire painful area once. When the area to be treated was too large, the application was then split several times. Each of these sub-applications was defined as a session. When all the sessions had been completed, another application could occur, constituting one or several sessions. Those sessions were usually programmed at most with a week interval. The evaluation of such split application was then programmed a month after all the sessions were completed.

### 2.3. Data Collection

The data were obtained from the ICO patients’ electronic medical records: DxCare© (Medasys, Clamart, France). This medical record included data about consultations, medication prescriptions, and hospitalizations. The data were transferred via a dedicated and protected database. The data were screened and collected by two different pain physicians. Nine different pain physicians participated in the data collection. This data collection lasted up to 22 December 2021 (last record of included patients at the time of data collection).

### 2.4. Evaluating the Primary Objectives

The efficacy evaluation was based on the Clinician’s Global Impression of Change (CGIC) during the successive HCCP applications, as detailed in Table 2. This CGIC was evaluated based on the occurrence of clinical criteria for pain relief in the follow-up consultations by the referent pain specialist. This was evaluated by two different pain physicians, with at least one specialist not being the one in charge of the patient during the treatment application. In case of conflicting analgesic effect evaluations by the two pain physicians, the most unfavorable evaluation was retained. In case of missing analgesic effect for one evaluation, the evaluation was fixed as “no effect” to limit modelling bias.

### 2.5. Statistical Analysis

Descriptive analysis was made with categorical variables summarized by numbers and percentages. Continuous variables were reported by mean and standard deviation or median and interquartile ranges. Sub-group comparisons were performed when appropriate for the purpose of hypothesis generation. Qualitative data were compared using a chi-square test or Fisher’s exact test when an expected value in a cell was lower than 5. Ordinal data were compared using a Cochran–Armitage or Kruskal–Wallis test when appropriate. Quantitative data were compared using a Mann–Whitney test or Kruskal–Wallis test when appropriate. All analyses were performed using R version 4.0.3 and their plugins R Commander (version 4.1.3) and R Studio (version 2022.07.0).

## 3. Results

### 3.1. Study Follow-Up

From 1 January 2014 to 22 December 2021, 987 patients received at least one HCCP application at the ICO Anaesthesiology and Pain Department, including 57 patients treated for CIPN. All 57 patients were included in this study. Overall, 457.3 patches were used, corresponding to 184 distinct applications. Each patient benefited from a mean of 3.2 (SD 3.3) applications, corresponding to a median of 2 (1;4) applications. For each session, patients had a mean of 1.7 (SD 0.8) patches applied, corresponding to a median of 2 (1;2) patches. This information is summarized in the flow chart (Figure 1).

### 3.2. Patient Characteristics

Of the 57 patients, 46 were female (80.7%), and 11 were male (19.3%). The median age was 59 (51; 69) years old. The most frequent comorbidity was hypertension, which was reported by 21 patients (36.8%). Patients mostly suffered from breast cancer (34 patients, 59.7%), followed by digestive cancer (8 patients, 14%), then lung and prostate cancer (respectively, 4 patients each, 7.0% each). The suspected chemotherapy was taxane alone for 30 patients (52.6%), platinum salts alone for 8 patients (14.0%), then taxane and platinum salts associated for 5 patients (12.5%), and taxane and immunotherapy for 5 patients (12.5%).

CIPN had lasted for less than a year for 25 patients (43.9%) at the time of the first HCCP application, one to five years for 23 patients (40.3%), and more than five years for 9 patients (15.8%). Medical records indicated that 39 (68.4%) patients had received at least one previous pain medication stopped before the HCCP application, consisting of antidepressants for 13 patients (22.8%), antiepileptics for 20 patients (32.1%), and opioids for 6 patients (10.5%). During the HCCP applications, 46 patients (80.7%) received at least one medication for neuropathic pain. HCCP applications were mostly used as the third or later treatment line in 28 patients (49.1%). They were used in second line for 21 patients (36.8%) and in first line for the last 8 patients (14.0%). Maximal pain before patch application was rated in 25 medical records, with a mean of 7.4 (SD 1.6) out of 10. All the data are summarized in Table 3.

### 3.3. Characteristics of HCCP Applications

Over the 296 sessions, 222 treated lower limbs only, 71 upper limbs only, and 3 sessions were realized over upper and lower limbs simultaneously, concerning 2 different patients. A total of 37 patients were treated on lower limbs only, 14 patients on upper limbs only, and 6 patients were treated on all four limbs. Each patient had 1 to 73.5 patches applied in total, with a median of 4.0 (2.0; 9.0) patches per patient. The median duration between two sessions was 53.5 days (8.25; 83.75).

### 3.4. HCCP Overall Efficacy

CGIC was available for 159 out of 184 applications. An important or complete analgesia was found in 61 applications (33.2%) concerning 30 patients (52.6%). Complete analgesia was found in 17 applications (9.2%) for 11 patients (19.3%). In total, 32 patients (53.1%) reported at least one application without any effect, and 14 patients (24.6%) never felt any analgesic effect after HCCP application. Over all the applications, 66 (35.9%) had no analgesic effect. The data are summarized in Table 4. The overall efficacy data are aggregated in Table 5.

### 3.5. HCCP Efficacy Depending on Several Factors

#### 3.5.1. Pain Duration before HCCP Application

When comparing to shorter duration, there was a significant increase in efficacy for pain duration of more than two years (*p* = 0.02) for all applications. Figure 2 highlights that complete or significant efficacy seemed to be more frequent the longer the pain had lasted.

#### 3.5.2. Responsible Chemotherapy

There was a significant difference in efficacy depending on the chemotherapy that induced the CIPN (*p* = 0.001). When comparing taxane-induced CIPN to others, where we expected a tendency for better HCCP efficacy, the difference was not significant (*p* = 0.2528), as shown in Figure 3, meaning that HCCP efficacy is the same for taxane-induced CIPN compared to others.

Platinum-salts-induced CIPN had a significantly worse response to HCCP application than other chemotherapies (*p* = 0.0238), as shown in Figure 4.

#### 3.5.3. Analgesic Treatment Line

There was a significant difference in efficacy depending on what treatment line the HCCP were used in (*p* = 0.003). When used in first line, the sample was too small to allow an effective comparison with other groups more accurate than the Cochran–Armitage test. However, when comparing HCCP use in third and second line of treatment, we found a significant difference in favor of second line (*p* = 0.018). The results are summarized in Figure 5.

#### 3.5.4. Total Number of HCCP Applications

HCCP application efficacy was significantly better for patients who received three or more applications compared to those who received fewer (*p* = 0.0334). Figure 6 summarizes HCCP efficacy depending on the total number of applications the patients received.

### 3.6. Time Lapse before HCCP Efficacy and Duration of the Pain Relief

The time lapse between the HCCP application and the onset of pain relief was informed in 36 applications out of 184, evaluated as less than a week for 26 applications out of 36 (72.2%) and less than four days for 16 applications (44.5%). The duration of the pain relief was indicated in 105 applications out of 184. For 52 (49.5%) of them, the duration was evaluated as continuous between two applications. For 19 (18%), the efficacy remained up to a month; for 45 (42.9%), it remained up to 3 months.

### 3.7. HCCP Tolerability

Overall, HCCP was well-tolerated. The most common adverse events were local reactions: 38 patients reported a local reaction at least once after HCCP application. Those local reactions were mostly a burning or painful sensation for 32 patients or erythema for 12 patients. Systemic adverse events were rare. There was no reported frostbite due to local cooling during or after the HCCP application. The occurrence of an adverse event seemed more frequent during the first application compared to the following ones, as shown in Figure 7, but the trend was not significant (*p* = 0.149).

HCCP tolerance data are summarized in Table 6.

### 3.8. End of Treatment and End of Study

The data were censored on 22 December 2021, while 15 patients were still being treated with HCCP applications for painful CIPN (26.3%). For the other patients, 8 (14.0%) stopped the treatment because they were no longer experiencing pain, 12 (21.1%) died before the end of the study, 6 (10.5%) stopped the applications at their own request, and 10 (17.5%) were lost to follow-up. All deaths were related to cancer.

The mean follow-up duration was 472 (SD 430.3) days with a median of 297 (154.8; 715) days.

## 4. Discussion

### 4.1. HCCP Efficacy

A significant (>50%) or complete pain relief in 43.9% of the patients was observed as well as in 33.2% of all the applications. HCCP thus seem to be an interesting treatment for chronicized painful CIPN, where only duloxetine had previously shown efficacy in 59% of the patients [22]. This corroborates the study by Le Marec et al., which found HCCP efficacy of 70% in 39% of patients using the BPI scale [34]. There is a significant difference in efficacy depending on the time between the occurrence of CIPN and HCCP application (*p* = 0.0005), which could mean that this treatment is more potent with time. However, this can be qualified by the natural evolution of CIPN, which tends to regress with time [6].

We identified a significant difference in efficacy depending on the chemotherapy responsible (*p* = 0.001) and further showed that the pain relief for CIPN related to platinum salts tended to be significantly worse than with other chemotherapies (*p* = 0.0238). In the meantime, we found a non-significant tendency for a better response with CIPN induced by taxanes (*p* = 0.2528). This could partly be explained by one of the taxane-linked CIPN induction mechanisms, with TRPV1 receptor overexpression in sensitive neuronal soma in the dorsal root ganglia through prolonged activation of glutamate receptors, inducing hyperalgesia [37,38]. Local HCCP applications may then desensitize those sensitive neurons upstream [27].

One important point shown by this study is the significant increase in efficacy in HCCP applications by repeating the applications, with significantly better efficacy for patients who benefited from three or more HCCP applications compared to those who received fewer (*p* = 0.0334). A similar potentiated efficacy has been shown in other forms of neuropathic pain regardless of the cause [39,40]. This supports the principle of applying at least three HCCPs before concluding that the efficacy of the treatment is insufficient.

### 4.2. HCCP Tolerability

The tolerability of HCCP applications was generally good. Although 66.6% patients reported a side effect at least once during application, this corresponded to 47.3% of the total applications, and we found a tendency for a decrease in the occurrence of adverse events in following applications although this was not significant (*p* = 0.149). The most frequent adverse event was the occurrence of a burning or pain sensation on the application area for 56.1% of the patients, corresponding to 41.4% of the applications. However, the pain sensation was usually well-tolerated, as it was evaluated as intense or very intense by 21% of the patients and only concerned 11.4% of the total applications and was evaluated without any premedication. All etiologies included, HCCP application in neuropathic pain has generally been labelled as having good tolerability [41].

### 4.3. CIPN in CERCAN Cohort and General Population

This study follows the constitution of the CERCAN cohort at the ICO, focusing on the efficacy of HCCP patches in cancer-induced neuropathic pain, with the first part focusing on breast cancer [36]. During that first study, 11.8% of the patients suffered from painful CIPN. In the current study, 58 of the 987 patients in the whole CERCAN cohort benefited from HCCP applications for painful neuropathy, i.e., 5.9%. This proportion is way below the 30% prevalence at three months but can be explained by the mean time lapse before HCCP treatment of the CIPN, which was 23.6 months, combined with the natural evolution of CIPN, which is mostly favorable. This time lapse is close to what can be seen in other publications [32]. More than 85% of the patients benefited from HCCP applications as the second or third line of pain treatment, which corresponds to the ESMO recommendations. The initial pain before HCCP application was evaluated as 7.4 (SD 1.6) out of 10 on a numeric scale, and the same levels are observed in the literature [33].

### 4.4. Why HCCP in Painful CIPN?

Many medications have been tested over time, mostly without any significant effect. No preventive treatments for the occurrence of CIPN seem efficient. Acetyl-L-carnitine has not shown any efficacy preventing the occurrence of CIPN and may even increase it [42,43]. No recommendation to prevent the occurrence of CIPN has been published. Suspected effective means of reducing their occurrence are to adjust chemotherapy doses and cool extremities using cryotherapy [44,45]. Topical menthol applications seem to be effective in early open studies [46]. Concerning non-pharmacological treatments, only physical exercise and functional training have proven effective to some extent [47]. With this limitation on the available treatments in painful CIPN, HCCP seems a very promising alternative.

### 4.5. Limitations and Strengths

This study presents some limitations. Due to its retrospective nature, there was a significant risk of missing data, which can mostly be found in the tolerability part of the study. However, data collection by two different investigators and systematic integration of the consultation reports into the electronic medical records of the patients limited this missing data phenomenon, most particularly in terms of efficacy evaluation. There is also a selection bias in the study population caused by the monocentric nature of this study. Breast cancer is over-represented (59.7% of the patients) due to the significant senological activity of the ICO comprehensive care center in Angers. This can also lead to overexposure to taxanes in the study population, often used as first line in breast cancer, compared to other chemotherapies. The naturally favorable evolution of CIPN must also be considered, as it can cause an interpretive bias on pain after 5 years of evolution [7]. However, this only concerns nine patients (15.8%) from the study population, which is a minority. Another interpretive bias is the variation in the concomitant analgesic treatments, which was not collected and may have acted on the overall evaluation of efficacy. Another limitation of this study is that pain scores before patch application were only available in less than half of the series. Finally, we can regret that the size variation in the painful area was not evaluated because the study was retrospective.

Nevertheless, there are few studies about this topic in the literature, and this is one of the largest cohorts studied as of today. Moreover, in the HCCP efficacy evaluation, to limit the data loss, two different pain specialists carried out a double screening, with no mutual consultation. When the two evaluations were conflicting, the most unfavorable one was retained. Further, to evaluate the primary objective, in cases where data was missing, the evaluation was given as “No effect”. This may have led to undervaluation of the efficacy of HCCP in painful CIPN.

## 5. Conclusions

This study demonstrates that HCCP could be a valuable therapeutic option in the treatment of painful CIPN due to the lack of effective systemic analgesic treatments available and the significance of their side effects. Local application of HCCP in painful CIPN can produce significant pain relief for patients, with usually suitable tolerability. The efficacy of the patches tends to increase with repeated applications. It is thus important to evaluate the efficacy of HCCP applications in painful CIPN prospectively compared to systemic duloxetine, the gold standard.

## Figures and Tables

**Figure 1 cancers-15-00349-f001:**
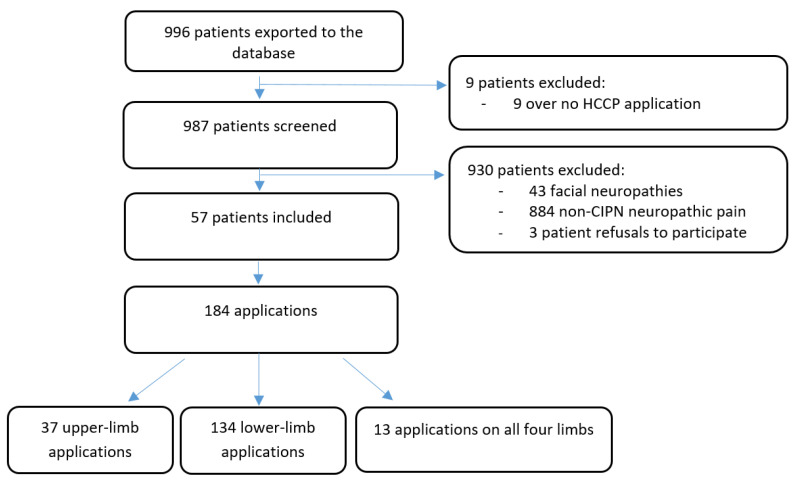
Flow Chart.

**Figure 2 cancers-15-00349-f002:**
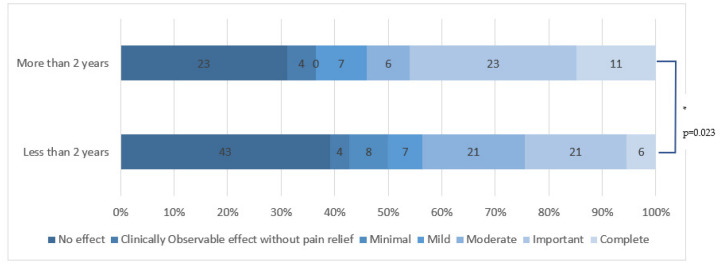
HCCP application efficacy depending on painful CIPN duration. * means that there is a statistically significant difference.

**Figure 3 cancers-15-00349-f003:**
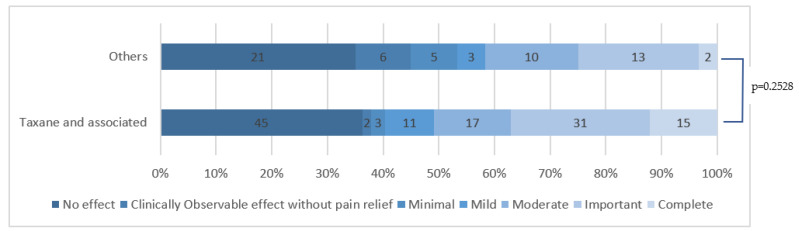
HCCP application efficacy in taxane-induced (and associated) CIPN compared to other chemotherapies.

**Figure 4 cancers-15-00349-f004:**
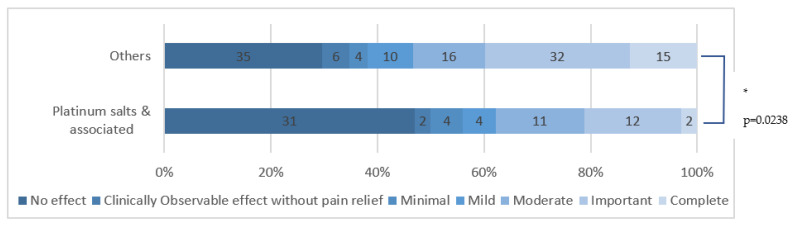
HCCP application efficacy in platinum-salts-induced (and associated) CIPN compared to other chemotherapies. * means that there is a statistically significant difference.

**Figure 5 cancers-15-00349-f005:**
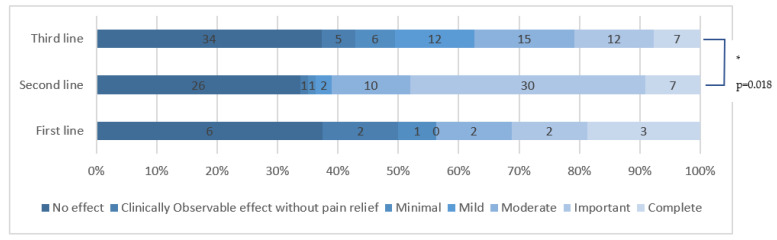
HCCP efficacy depending on the analgesic treatment line. * means that there is a statistically significant difference.

**Figure 6 cancers-15-00349-f006:**
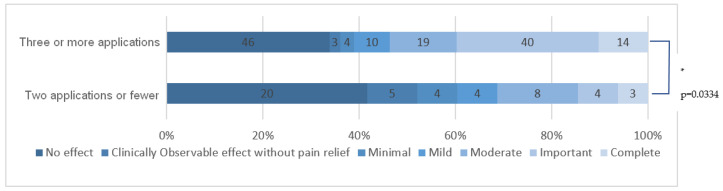
HCCP efficacy for patients who received three or more applications compared to those who received fewer. * means that there is a statistically significant difference.

**Figure 7 cancers-15-00349-f007:**
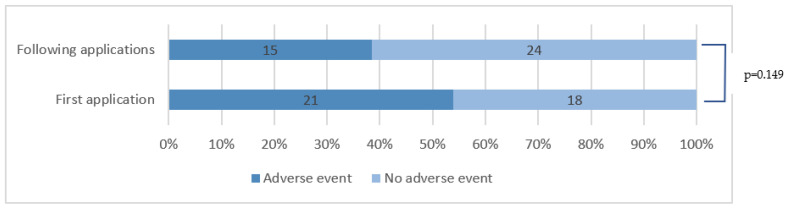
Adverse effect occurrence during the first HCCP application compared to subsequent applications.

**Table 1 cancers-15-00349-t001:** Current evidence concerning capsaicin patch in CIPN population.

Study	Publication Year	Type of Study	Number of Patients	Responsible Chemotherapy	HCCP Application Duration	Results
Anand et al. [32]	2019	Prospective	16	Platinum salts, taxanes, and/or bortezomib	30 min	Significant −1.27 (−0.24; −2.3) diminution in NRS scores at 12 weeks
Filipczak-Bryniarska et al. [33]	2017	Prospective	18	Platinum salts	As long as possible	Pain reduction of 84% and 97% in high and low platinum sensitivity groups respectively at 12 weeks
Le Marec et al. [34]	2016 (abstract)	Retrospective	28	Platinum salts, taxanes, bortezomib, thalidomide, or alkeran	30 min	Pain scores reduced by >50% in 21 patients (75%) at 6 months
Ramnarine et al. [35]	2016 (abstract)	Retrospective	19	N/A	N/A	7 responders (37%) at 4 weeks corresponding to a 30% pain scores reduction and 7 responders (50%) at 12 weeks.

**Table 2 cancers-15-00349-t002:** Clinical Global Impression of Change (CGIC).

Analgesic Effect	Definition
No effect	No change
Clinically observable effect but no pain relief	Reduction in pain area **OR** decreased intensity of allodynia or hyperalgia
Minimal effect	2-point (NRS) or 1-category (VRS) decrease in mean pain intensity **AND/OR** 2-point (NRS) or 1-category (VRS) decrease in maximum pain intensity
Mild effect	2-point (NRS) or 1-category (VRS) decrease in mean pain intensity **AND** 2-point (NRS) or 1-category (VRS) decrease in maximum pain intensity **AND changes in one or more of the following criteria:** 1/ Decrease in pain flare frequency (but at least 2/day) 2/ Decrease in sleep interference score (but at least 1 awake/night) 3/ Slight decrease in daily activities interference score 4/ Slight decrease in evoked pain reported by the patient 5/ 50% decrease in the daily dose of at least one neuropathic pain medication 6/ Breakthrough pain analgesic dose reduction or cessation
Moderate effect	30% to 50% (NRS) or 2-category (VRS) decrease in mean pain intensity: **OR** 30% to 50% (NRS) or 2-category (VRS) decrease in maximum pain intensity **AND changes in one or more of the following criteria:** 1/ At least 50% decrease in pain flare frequency 2/ No more sleep interference (or rare) 3/ At least 50% decrease in daily activities interference score 4/ Cessation of at least one neuropathic pain medication OR at least 50% decrease in the daily dose of two neuropathic pain medications
Important effect	30% to 50% (NRS) or 2-category (VRS) decrease in mean pain intensity: **OR** 30% to 50% (NRS) or 2-category (VRS) decrease in maximum pain intensity **AND changes in two of the following criteria:** 1/ At least 50% decrease in pain flare frequency 2/ No more sleep interference (or rare) 3/ At least 50% decrease in daily activities interference score 4/ Cessation of at least one neuropathic pain medication OR at least 50% decrease in the daily dose of two neuropathic pain medications
Complete effect	No pain flares (or <1/week max) Usual pain: absent No pain interference on sleep (or ≤1/week) No more neuropathic pain medication

**Table 3 cancers-15-00349-t003:** Patients’ and HCCP Treatment Characteristics.

Baseline Characteristics (N = 57)	N (%) or Mean (SD)
**Sex**	
Female	46 (80.7)
Male	11 (19.3)
**Cancer Localization**	
Prostate	4 (7.0)
Lung	4 (7.0)
Oligodendroglioma	1 (1.8)
ORL	1 (1.8)
Uterus	1 (1.8)
Acute leukemia	1 (1.8)
Myeloma	1 (1.8)
Testis	34 (59.7)
Breast	8 (14.0)
Digestive cancer	
**Comorbidity**	
Hypertension	21 (36.8)
Infarct	2 (3.51)
Pulmonary embolism	1 (1.75)
Stroke	2 (3.51)
**Age** (median) (IQR)	59 (51; 69)
**Suspected chemotherapy**	
Taxane alone	30 (52.6)
Platinum salts	8 (14.0)
Taxane + platinum salts	5 (12.3)
Taxane + immunotherapy	5 (12.3)
Platinum salts + immunotherapy	2 (3.5)
Taxane + Platinum salts + immunotherapy	2 (3.5)
Immunotherapy	1 (1.8)
Vincristine	2 (3.5)
Thalidomide + lenalidomide	1 (1.8)
Thalidomide + lenalidomide + bortezomib	1 (1.8)
**Duration of CIPN before capsaicin use**	
<1 year	25 (43.9)
1–5 years	23 (40.3)
>5 years	9 (15.8)
**Previous pain medication**	
At least one previous (stopped before inclusion)	39 (68.4)
At least one antidepressant	13 (22.8)
At least one antiepileptic	20 (35.1)
At least one opioid	6 (10.5)
**Ongoing analgesic medication for neuropathic pain**	
At least one ongoing analgesic medication	46 (80.7)
**Treatment line for CIPN**	
First line	8 (14.0)
Second line	21 (36.8)
Third line (or more)	28 (49.1)

**Table 4 cancers-15-00349-t004:** Efficacy evaluation of HCCP treatment with CGIC scores per patient and per application.

Analgesic Effect	Per Patient (N = 57)		Per Application (N = 184)	
	N (%)		N (%)	
Complete	11 (19.3)		17 (9.2)	
Important	14 (24.6)		44 (23.9)	
Moderate	9 (15.8)		27 (14.7)	
Mild	3 (5.3)		14 (7.6)	
Minimal	3 (5.3)		8 (4.3)	
Clinically observable effect without pain relief	3 (5.3)		8 (4.3)	
No effect	14 (24.6)		66 (35.6)	

Notes: Analgesic effect was determined by two pain specialists, with at least one of them not the treating physician. It was based on the electronic medical file of the included patients (184 applications). In case of conflicting analgesic effect evaluations by the two pain physicians, the most unfavorable evaluation was retained. If only one evaluator reported an analgesic effect, it was retained. If no evaluation was found by either evaluator, the evaluation was labelled “No effect”. For patient evaluation, the analgesic effect was determined by the maximal effect reported after any of the applications when more than one application was performed.

**Table 5 cancers-15-00349-t005:** Efficacy evaluation of HCCP treatment with CGIC scores depending on pain duration, responsible chemotherapy, treatment line, and ongoing pain medications.

Characteristics		Application (N = 184)	CGIC							*p*-Value
			No effect	Clin. E	Min E.	Mild E.	Mod. E	Imp. E	Comp. E	
			n (%)	n (%)	n (%)	n (%)	n (%)	n (%)	n (%)	
Pain duration	<1 year	52	16 (30.8)	2 (3.8)	4 (7.7)	3 (5.8)	16 (30.8)	5 (9.6)	6 (11.5)	0.0005 *
	1–2 years	58	27 (46.6)	2 (3.4)	4 (6.7)	4 (6.7)	5 (8.6)	16 (27.6)	0 (0)	
	3–4 years	32	12 (37.5)	1 (3.1)	0 (0)	5 (15.6)	3 (9.4)	9 (28.1)	2 (6.3)	
	>5 years	42	11 (26.2)	3 (7.1)	0 (0)	2 (4.8)	3 (7.1)	14 (33.3)	9 (21.4)	
Responsible chemotherapy	T	94	27 (28.7)	2 (2.1)	1 (1.1)	9 (9.6)	12 (12.8)	31 (33.0)	12 (12.8)	0.001 *
	P	45	16 (35.6)	2 (4.4)	4 (8.9)	2 (4.4)	7 (15.6)	12 (26.7)	2 (4.4)	
	T + P	8	5 (62.5)	0 (0)	0 (0)	0 (0)	3 (37.5)	0 (0)	0 (0)	
	T + I	13	5 (38.5)	0 (0)	2 (15.4)	1 (7.7)	2 (15.4)	0 (0)	3 (23.1)	
	P + I	4	2 (50.0)	0 (0)	0 (0)	1 (25.0)	1 (25.0)	0 (0)	0 (0)	
	T + P + I	9	8 (88.9)	0 (0)	0 (0)	1 (11.1)	0 (0)	0 (0)	0 (0)	
	I	1	0 (0)	0 (0)	1 (100)	0 (0)	0 (0)	0 (0)	0 (0)	
	Vincristine	4	2 (50.0)	1 (25.0)	0 (0)	0 (0)	1 (25.0)	0 (0)	0 (0)	
	Th + L	2	0 (0)	2 (100)	0 (0)	0 (0)	0 (0)	0 (0)	0 (0)	
	Th + L + B	4	1 (25.0)	1 (25.0)	0 (0)	0 (0)	1 (25.0)	1 (25.0)	0 (0)	
Treatment line	First line	16	6 (37.5)	2 (12.5)	1 (6.3)	0 (0)	2 (12.5)	2 (12.5)	3 (18.8)	0.003 *
	Second line	77	26 (33.8)	1 (1.3)	1 (1.3)	2 (2.6)	10 (13.0)	30 (39.0)	7 (9.1)	
	Third line	91	34 (37.4)	5 (5.5)	6 (6.6)	12 (13.2)	15 (16.5)	12 (13.2)	7 (7.7)	
Ongoing pain medication	No	21	5 (23.8)	0 (0)	1 (4.8)	3 (14.3)	0 (0)	8 (38.1)	4 (19.0)	0.046 *
	Yes	163	61 (37.4)	8 (4.9)	7 (4.3)	11 (6.7)	27 (16.6)	36 (22.1)	13 (8.0)	
Antidepressants	No	94	36 (38.3)	4 (4.3)	4 (4.3)	6 (6.4)	14 (14.9)	22 (23.4)	8 (8.5)	0.7431
	Yes	90	30 (33.3)	4 (4.4)	4 (4.4)	8 (8.9)	13 (14.4)	22 (24.4)	9 (10.0)	
Antiepileptics	No	83	30 (36.1)	4 (4.8)	2 (2.4)	9 (10.8)	6 (7.2)	27 (32.5)	5 (6.0)	0.0005 *
	Yes	101	36 (35.6)	4 (4.0)	6 (5.9)	5 (5.0)	21 (20.8)	17 (16.8)	12 (11.9)	
Opioids	No	151	56 (37.1)	5 (3.3)	7 (4.6)	13 (8.6)	21 (13.9)	33 (21.9)	16 (10.6)	0.08
	Yes	33	10 (30.3)	3 (9.1)	1 (3.0)	1 (3.0)	6 (18.2)	11 (33.3)	1 (3.0)	

Notes: T, taxanes; P, platinum salts; I, immunotherapy; Th, thalidomide; L, lenalidomide; B, bortezomib. CGIC score: Clin. E, clinically observable effect without pain relief; Min. E, minimal effect; Mild E, mild effect; Mod. E, moderate effect; Imp. E, important effect; Comp. E, complete effect. * means that there is a statistically significant difference between the lines of a category.

**Table 6 cancers-15-00349-t006:** HCCP tolerability per patient, application, and session.

Adverse Event	Per Patient n = 57 (%)	Per Application n = 184 (%)	Per Session n = 296 (%)
Local reaction	38 (66.6%)	87 (47.3%)	118 (39.9%)
Burning or painful sensation	32 (56.1%)	77 (41.4%)	107 (36.1%)
Erythema	12 (21.1%)	17 (9.2%)	19 (6.4%)
Systemic reaction	2 (3.5%)	2 (1.1%)	2 (0.6%)
Hypertensive crisis	1 (1.7%)	1 (0.5%)	1 (0.3%)
Vasovagal syncope	1 (1.7%)	1 (0.5%)	1(0.3%)
**Local pain during the application**			
Very intense	4 (7.0%)	4 (2.2%)	5 (1.7%)
Intense	8 (14.0%)	17 (9.2%)	21 (7.1%)
Moderate	20 (35.1%)	48 (26.1%)	64 (21.2%)
Light	9 (15.8%)	57 (31%)	93 (31.2%)
No data	16 (28.1%)	58 (31.5%)	113 (38.2%)
**Local pain after discharge**			
Very intense	8 (14.0%)	8 (4.3%)	9 (3.0%)
Intense	20 (35.0%)	29 (15.8%)	35 (11.8%)
Moderate	6 (10.5%)	21 (11.4%)	26 (8.8%)
Light	6 (10.5%)	26 (14.1%)	31 (10.4%)
No pain	1 (1.8%)	5 (2.7%)	8 (2.7%)
No data	11 (19.3%)	95 (51.6%)	187 (63.2%)

## Data Availability

The data that support the findings of this study are available from F. Bienfait, the corresponding author, but restrictions apply to the availability of these data, which were used under license for the current study and so are not publicly available. Data are, however, available from the authors upon reasonable request and with permission of F. Bienfait, the corresponding author.

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
