# Peer review of "Evaluation of 8% Capsaicin Patches in Chemotherapy-Induced Peripheral Neuropathy: A Retrospective Study in a Comprehensive Cancer Center"

_cancers, 2023, doi:10.3390/cancers15020349_

Round 1

Reviewer 1 Report

The manuscript describes and analyze the results of a retrospective clinical study focusing on the analgesic efficacy of high concentration capsaicin patch for patients suffering from peripheral neuropathy due to anticancer chemotherapy.

As several patients experience neuropathic pain, that is difficult to treat, any sound new approach is welcome to be tested. Therefore, the current manuscript deserves attention and may provide significant contribution to new possible treatment options to postchemotherapeutic neuropathy. However, there are some major concerns that must be answered before recommending the paper for publishing:

1. The main concern is raised by the interpretation of analgesic effect and the scoring of the Clinical Global Impression of Change (CGIC):

a, It is not understood why the decrease in allodynia or hyperalgesia is not considered analgesic effect. As it is found, there was no allodynia or hyperalgesia measuring score applied.

b, Up to the score of the "Mild effect" the evaluation properly includes the absolute decrease in the Numeric Rating Scale (NRS), meaning that how many points decrease could be observed. From "Moderate effect" the evaluation suddenly switches to percentage decrease in NRS, this way it is not known what the initial NRS score was and how many points it decreased after applying the patch. For example, if NRS decrease is 50%, it may be from score 8 to 4, or from score 4 to 2, that is not the same. The reviewer suggest reconsidering the whole evaluation and using the absolute score decrease instead of percentage change.

2. Capsaicin patch is indicated for peripheral neuropathy, but its efficacy was proven only for postherpetic or HIV caused neuralgia/neuropathy and diabetic neuropathy (The authors properly cite the EPAR of Qutenza). Even if ESMO Guideline recommends capsaicin patch for chemotherapy induced neuropathy, detailed application protocol must be provided since this type of neuropathy lacks accepted treatment protocol. The precise application, e.g duration of patch use, is not known from the manuscript. If the application protocol was the same as in the Summary of Product Characteristics for licensed indications, it must be mentioned in the manuscript.

3. The authors describe in the Methods section that in case of a large area, patches were applied in a timely splitted manner. How can the authors explain any analgesic effect in this case, since the non-treated area might have remained as painful as before. It must be clarified in a revised version of the paper.

There are some unclear issues in the tables and figures:

1. In Table 4 what does N represent? For example, summing up the pain duration N values, it does give the number of distinct applications (184) but it must be specified in the table explanation.

2. When Figure 2 is explained, the authors claim that there was a significant difference among the groups of different duration of neuropathic pain. However, there is no significance level is shown in the Figure, only in the paragraph before the Figure, and it is not specified what was exactly compared. Overall pain relief pattern or only the responders?

3. The same concern above applies to Figure 3.

4. In Figure 3 the authors claim a tendency to efficacy in taxane-induced neuropathy. However, if the significance level does not reach the predefined value that is considered the level of significance, one cannot say that the treatment is efficacious. It must be noted that capsaicin was not effective against taxane-induced neuropathy in the current study.

5. The previous two concerns apply to Figure 5.

The Discussion section mainly repeats the Results. The authors claim that only duloxetine was effective that is hard to believe, more focus should be put on the other measures, such as antiepileptics or opioids, referring to papers that found or didn't find effects of the above-mentioned analgesics. A lot of papers are cited that try to elucidate the mechanisms of neuropathic pain, but only taxane-induced neuropathy mechanism is mentioned. It would be supported if a short summary of different mechanisms were described in the Discussion or in the Introduction.

As a conclusion, the manuscript deals with an important issue, therefore it should not be rejected, after a major revision as suggested above, it could be accepted for publication.

Author Response

Thank you for your comments and advice.

Please see the response as attached file.

Reviewer 2 Report

Review manuscript: cancers-2082381

Title: Evaluation of 8% Capsaicin Patches in Chemotherapy-Induced Peripheral
Neuropathy

This is a retrospective single center subanalysis of a larger study, currently including 57 patients with chemotherapy induced neuropathy (I guess, it is not well defined) in whom the efficacy of the 8% capsaicin patch is assessed in a total of 184 applications based on reviewing the medical records and rating retrospectively the physician´s impression of change during patch application. Overall, it represents a methodologically weak study financed by the industry where the utilization of a subjective rating retrospectively is a major limitation, and with also a bad structured redaction of main parts, what precludes my support to be considered per publication in Q1 journal as Cancers.

Major comments:

In the manuscript, is not clearly defined the population. Which is the definition that authors use for CIPN (chemotherapy induced peripheral neuropathy or chemotherapy-induced neuropathic pain? (line 14, simple summary). This should be clearly clarified.

The main objective of the study is well exposed in line 145. However, the scale used to assess the efficacy is a subjective retrospectively scale rated by physicians who review the medical records of the patients, what implies a major limitation in the study. CGIC is inferred and main comparisons are done between those having complete or > 50% improvement (“important”) vs the other. Noteworthy, no specific information on the criteria used to define efficacy (complete? Important? Moderate?..) is stated in the text. Importantly, no data on patient´s reported outcomes during patch application neither neurological background of the symptoms are available. Severity of pain before patch application was only available in less than half of series, what represents an important limitation to be highlighted.

Manuscript redaction needs to be much more improved and summarized. Less paragraphs must be used in the redaction, with more sentences in every paragraph. Ideas should be introduced or discussed together. In the results, those data showed in the tables don´t have to be also written in the text. Many information is redundant. Text should be for emphazise data showed on the table. The abstract should contain meaningful and specific information. I would consider the help of a medical writer to improve the text.

Detailed comments:

Line 14: CIPN definitions should be clarified. Currently it does not have meaning: “ chemotherapy-induced neurpathic pain (CIPN) are often painful…”.. Always according your sentence. Please check this important aspect of the term.

Lines 39 and 40: The sentences should include the chemotherapy agent or the treatment line that become significant, to be more specific and informative.

Introduction: Is too long, it should be summarized including two or maximum three paragraphs to introduce the problem of CIPN, and the current data on management of neuropathic pain in this setting and specific data on capsaicin . For example lines 91-96 are unnecessary. Importantly, several incorrect and confounding information is present and should be corrected. For example, in Line 70: coasting effect is not what they say: This means the new occurrence or worsening of ongoing neuropathy up to three months after finishing chemotherapy. Additionally, damage in the neuronal soma of peripheral neuron can not induce motor neuropathy and alterations in the motor strength, as suggested by the authors in lines 77-79. Rates of long-term CIPN depend on the agent ( sentence in line 67-68 needs reference). A summarizing table with the current evidence on capsaicin patch in CIPN population should be helpful for the reader to understand the available (lack) of evidence that supports the need of this study.

Lines 141-143: It is not clear if patients included in this study have been previously included in other publication.

Lines 151-160: It should be one paragraph. Use term exclusion criteria ( instead of non-inclusion).

Table 1: It should be improved to be more readable at a glance.

Figure 1, flow chart: data on number of patches and sessions should be deleted. Site of applications could be added.

Line 245 ; CGIC was available for 159 out of 184 applications. This should include the %. But, how many applications are included in the analysis 159 or 184?? Because CGIC is the main objective of the study…

Table 3: Column with 25 (43,9) and 61 (33.2) . what represents?

Table 4: CGIC is classified with numbers? Which number represent every category? P values in the last column… what they refer to? Which comparisons? The term suspected chemotherapy should be reconsidered… suspected???; Taxanes should be T in all lines of the table. B means bortezomib ( not bortezomide).

Figures 2 to 7: The percentages should be specified what they refer to.  I would suggest to show the comparisons between the two groups (complete + important vs other) or better specifiy the differences in the text.

Discussion: It requires improvement in the form and the content. i.e, the first paragraph should include the main result of the study. Line 367 discusses regarding pain distribution when no information has been previously provided in the text. Discussion have to discuss the  results previously showed, with the previous data reported in the literature.

Author Response

(The authors gave the same response as above.)

Round 2

Reviewer 2 Report

The manuscript is now much more improved. Some minor questions should be corrected.

Abstract: I would suggest to change " we found a significant decrease in efficacy for platinum salts" by " we found less efficacy for platinum..." to be more clear.

These two sentences from the abstract seem opposite: The efficacy was significatively  higher when HCCP was used in second line compared to third line (p=0.018). The efficacy of HCCP was significatively higher starting the third application (p=0.0334). Please clarify.

Table 1: Prospective in Anand study should be in capital letter, same as other

Point 4.3 in the discussion should be deleted if it is not based to discuss the results of the study. The discussion section is for discussing reults. If not, should be part of introduction.

Author Response

Please find the last response to your comments in the attached file.
